DOI: 10.1038/s41467-018-05708-1　　OPEN

# A balance between aerodynamic and olfactory performance during flight in *Drosophila*

Chengyu Li[1,4], Haibo Dong[2] & Kai Zhao[1,3]

The ability to track odor plumes to their source (food, mate, etc.) is key to the survival of many insects. During this odor-guided navigation, flapping wings could actively draw odorants to the antennae to enhance olfactory sensitivity, but it is unclear if improving olfactory function comes at a cost to aerodynamic performance. Here, we computationally quantify the odor plume features around a fruit fly in forward flight and confirm that the antenna is well positioned to receive a significant increase of odor mass flux (peak 1.8 times), induced by wing flapping, vertically from below the body but not horizontally. This anisotropic odor spatial sampling may have important implications for behavior and the algorithm during plume tracking. Further analysis also suggests that, because both aerodynamic and olfactory functions are indispensable during odor-guided navigation, the wing shape and size may be a balance between the two functions.

[1] Department of Otolaryngology, The Ohio State University, Columbus, OH 43210, USA. [2] Department of Mechanical and Aerospace Engineering, University of Virginia, Charlottesville, VA 22903, USA. [3] Department of Biomedical Engineering, The Ohio State University, Columbus, OH 43210, USA. [4]Present address: Department of Mechanical Engineering, Villanova University, Villanova, PA 19085, USA. Correspondence and requests for materials should be addressed to K.Z. (email: zhao.1949@osu.edu)

nsects have remarkable flying and odor-tracking capabilities[1–3] that have captivated the interest of naturalists and biologists for centuries. For example, a male moth can track its female counterpart from miles away via pheromone detection[4,5]. Insects possess a sophisticated olfactory system that is extremely sensitive to a great number of volatile chemicals, with its total detection capacity that has never been fully cataloged[6,7]. But then, how are olfactory detection and tracking affected by wing flapping during the flight, which inevitably perturbs the incoming odor plume? One widely held hypothesis is that flapping wings may actively draw odor plumes to the primary olfactory sensory organ, the antennae—an action analogous to "sniffing" in mammals, and that wing beating may be a critical part of active olfactory sampling for insects[8,9]. For example, a silkworm with its wings removed is unable to track odor plumes, even though it tracks plumes while walking[10]. Experimental measurements using hot-wire anemometers showed that the induced airflow generated by the wings varied with wingbeat frequency, which may alter olfactory stimuli to the sensory organs[11]. Yet, there is a lack of quantitative details and confirmation on how flapping wings actively draw odor plumes to the antennae.

On the other hand, through million years of evolution, insects have developed superior wing designs and complex flying mechanisms to enhance their aerodynamic performance[12–14]. Wing flapping back and forth generates a tornado-shaped leading-edge vortex on its top surface that can provide almost twofold lift compared with static wings[15,16]. Unsteady wing flapping motion can augment lift through delayed stall when the wing sweeps through the air with high angle of attack during the translational phase. Rapid wing reversal can capture its own wake and thus further enhance lift. Besides these three most common unsteady aerodynamic mechanisms[13] (leading-edge vortex, delayed stall, and wake capture), many insects also apply other unique aerodynamic mechanisms to further increase lift. For example, the slender wings of mosquitoes can take advantage of the trailing-edge vortex[14]. For insects like butterflies, their wings may interact with each other to enhance force generation through clap-and-fling and clap-and-peel aerodynamics[17,18]. For insects with wider body shape, such as cicadas, wing–body interaction mechanisms may play a critical role to enhance lift generation[19].

Despite our improved understanding of the aerodynamics of insect flight, it is still unclear whether the need to maintain high aerodynamic efficiency is in conflict with the need to draw more odors to the antennae. An alternative hypothesis that may have been overlooked by the scientific community is that the antennae may be located on the head precisely to avoid the disruption of the odor plume structure, which may contain localization information[20,21]. Imagine a speed boat skimming across a calm lake; its bow is always hitting the calm water, ahead of the wake and the turbulence generated by the propeller—if the insect indeed takes advantage of air perturbation generated by wing flapping, why are its antennae not located on its body or on its tail? Olfactory sensilla are located predominantly on the antennae and maxillary pulp, both near the tip of the head, potentially avoiding the disturbance created by wing flapping. Ultimately, since both aerodynamic and olfactory functions are indispensable during an odor-guided navigation, there has to be mechanism to balance both its aerodynamic and olfactory needs.

Here, we utilized an in-house high-fidelity computational fluid dynamics solver to simulate a fruit fly in forward flight motion. We quantified the odor mass flux around the insect body and visualized the odor plume structures through Lagrangian tracking method. The aerodynamic performance and vortex structures were also evaluated. The present effort explores the role of flapping wings in enhancing the olfactory stimulus and offers new insights into key regions of flapping wings that may differentially impact aerodynamic forces and antenna odor mass fluxes during odor-tracking flight.

## Results

**Modeling a fruit fly in forward flight**. We designed a "numerical wind tunnel" and simulated a morphological-accurate fruit fly model (Fig. 1a, b) in forward flight. Our simulated fruit fly is prescribed with realistic flapping kinematics (Supplementary Fig. 1) according to the literature[22,23] at a frequency of 213 Hz and a forward speed of 0.94 m/s, with a corresponding Reynolds number of 173, which describes the ratio of inertial to viscous forces in a fluid; and reduced frequency ($k$) of 0.65, which is the ratio of wing-tip velocity over forward velocity (see Methods and Supplementary Fig. 2). A normalized uniform pseudo-odor was released from the upstream inlet (see methods). This pseudo-odor can represent most natural odors in the environment, whose diffusivities in the air are generally quite low, ranging from $10^{-1}$ to $10^{-2}$ cm²/s. Since the convective odor transport due to air movement dominates the system (Peclet number $10^2$–$10^3$), the odor diffusion was ignored. Utilizing an in-house direct numerical simulation solver[24], we simulated the unsteady aerodynamics of the forward-flying fruit fly (Fig. 1b) and quantified its associated odor plume structures (Fig. 1c–e).

**Wing flapping enhances odor mass flux to antenna**. Our simulation confirmed that the flapping locomotion indeed enhanced the odor mass flux over its antennae (by ~1.8 times at its peak value, Fig. 1d position i). Surprisingly, odor flux along the fruit fly body and tail locations has lower peak intensity than at the antenna and is more chaotic due to the wake generated by the flapping wings (Fig. 1d and Supplementary Fig. 3). This finding confirmed that the conventional wisdom[25] is correct: during forward flight, the antennae are well positioned to receive significantly increased odor mass flux while avoiding significant air disturbance compared to other locations along the body. The mechanism for this enhanced odor mass flux to the antennae, as we observed, consists of two stages: trapping and flicking upward. In Fig. 1e and Supplementary Movie 1, the odor plume structure is visualized by odor particle tracing. The colors of the particles indicate different releasing locations. Without flapping motion, only a narrow jet of particles (the yellow particles) that are directly in the path of the antennae will pass over the antennae. With wing flapping, during the downstroke (Fig. 1e; $t/T = 6.75$ and $t/T = 7.00$), the wings push and trap odorous air below the body, preventing it from escaping downstream. Once the wings start the transition to upstroke (Fig. 1e; $t/T = 7.25$), the wide trailing edges close to the wing root rotate and flick the trapped odorous air (green and cyan particles) upward toward the antennae (Supplementary Movie 1, 00:38–01:15). The peak odor mass flux occurs not during upstroke or downstroke but, rather, during this wing transitional phase. This phase-locked odor mass flux within the wing-flapping cycle may be utilized by the olfactory system to enhance odor detection through potential neural connections to the motor centers[2,8].

**Effects of higher flapping frequencies**. To further explore the effects of flapping wings, different flapping frequencies ($k = 0.33$–$1.30$) were also simulated (Fig. 2). Figure 2a shows the top view of wake topology, using Q-criterion and color coded by the normalized pressure. In general, the wake is dominated by a chain of vortex loops behind each wing and periodically sheds off at the wing reversal points of the wing-beat cycle. Wing flapping induces a strong air vortex over the head (Fig. 2b) that intensifies with higher reduced frequency, which is the main driver of the increased odor mass flux to the antenna region, and potentially to

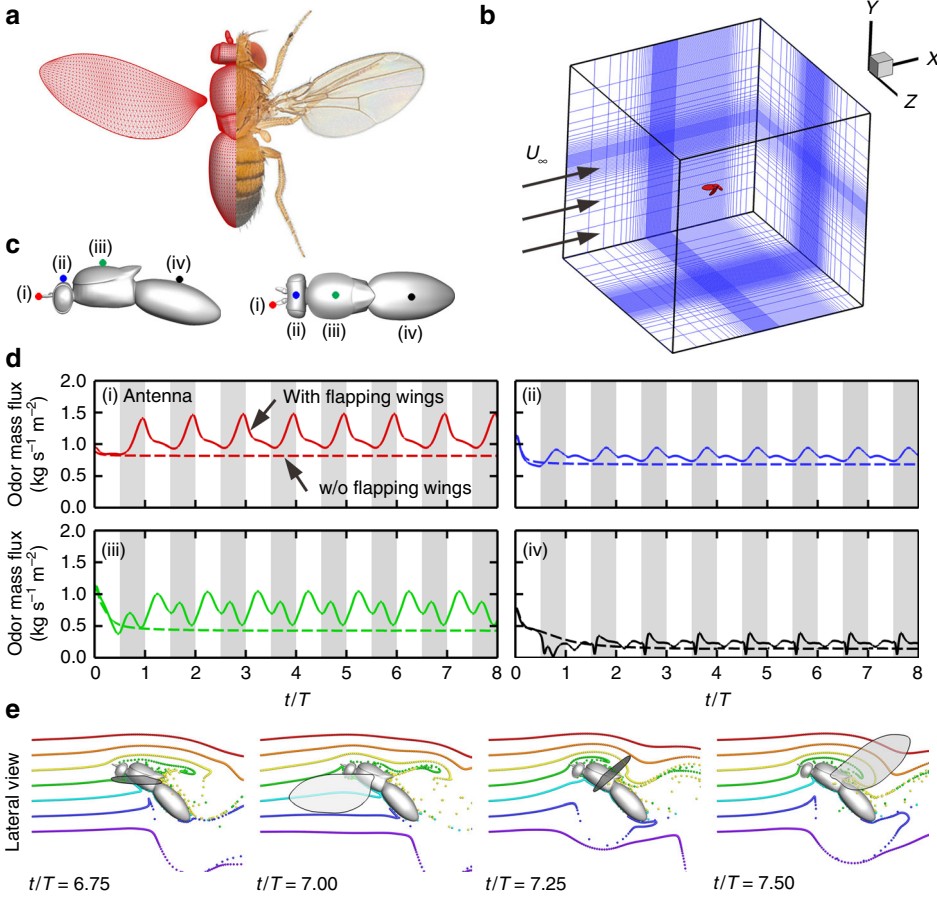

**Fig. 1** Modeling of fruit fly in forward flight. **a** The fruit fly *D. melanogaster*, with the computational model on the left half. High-density surface mesh with approximately 29,000 and 5000 triangular elements was used to define the body and each wing, respectively. The *Drosophila* image is credited to Tim Weil and Anna York-Andersen from Weil Lab at the University of Cambridge under creative common license. **b** Simulation setup is mimicking a fruit fly flying forward at a speed of 0.94 m/s and flapping frequency of 213 Hz (reduce frequency $k = 0.65$). The simulation has ~10 million computational grids. **c** The probe location for measuring the odor mass flux around fruit fly body. **d** The odor mass flux at the antenna (i) and different locations (ii–iv) around the fruit fly body (Fig. 1c). The shaded areas represent downstrokes. The dashed lines indicate the odor mass flux without wings flapping. The antenna is well positioned to receive significant increase of odor mass flux while avoiding significant turbulence compared to other locations along the body (see more in Supplementary Fig. 3). **e** Lateral view of odor particle tracers at various time points. The colors of the particles indicate different release locations. Left to right: middle downstroke ($t/T = 6.75$); supination ($t/T = 7.00$); middle upstroke ($t/T = 7.25$); pronation ($t/T = 7.50$). During the downstroke ($t/T = 0.65–7.00$), the flapping wing pushes and traps odorous air below the body, preventing it from escaping downstream. Once the wings start to reverse and flap upward ($t/T = 7.25–7.50$), the wide trailing edges close to the wing root rotate and flick the trapped odorous air upward toward the antennae. The peak odor mass flux at the antenna (Fig. 1d–i) occurs not during upstroke or downstroke but, rather, during this wing transition phase

the maxillary palp as well. Correspondingly, antenna odor mass flux increases significantly with higher flapping frequency (Supplementary Movie 1, 01:16–01:53), mostly because particles farther below the body (blue and purple) are also being perturbed and pushed up over the antenna region (Fig. 2c, $k = 1.30$). In another sense, the increased odor mass flux with wing flapping is the result of broader spatial sampling range. However, this spatial sampling is mostly limited vertically to below the body. In the horizontal plane (Fig. 2d; Supplementary Movie 1, 01:54–02:31), only a narrow stream of odor particles that is in the direct path of the body center can pass through the antennae, regardless how fast the wing flaps. The anisotropic spatial sampling ranges suggest that insects may have better capability to sample and detect odor plumes coming from below their body owing to their wing flapping. This may have implications in the behavior of plume tracking of most insects, which often consist of two distinct phases: surging upwind toward an odor source and zigzagging cross-wind (casting), which is triggered by loss of the odor plume[21]. Behaviorally, the zigzagging occurs more often horizontally than vertically[26], potentially because insects are able to

sample a wider spatial range in the vertical direction by wing flapping. Thus, horizontal casting is more essential to search and locate lost plumes. If we can speculate further, it might also be a reason for horizontally oriented antennae in moths (and many others), to potentially compensate for the lateral sampling range. These hypotheses, albeit speculative but amenable to future experimental investigation, may lead to further insights into insect odor-tracking behavior and algorithms.

**Balance between aerodynamics and olfaction.** Since both aerodynamic and olfactory functions are indispensable during odor-guided navigation, we set out to understand how insect wings achieve a balance between these seemingly conflicting roles. As shown in Fig. 2e, most of the lift force is produced by the wings during the downstroke and peaks near the mid-downstroke, corroborating previous study[22], but in a very different phase from odor mass flux peaks (Fig. 2f). The cycle-averaged lift coefficient and odor mass flux over the antennae obtained from the simulations are summarized in Table 1, and they do not follow the

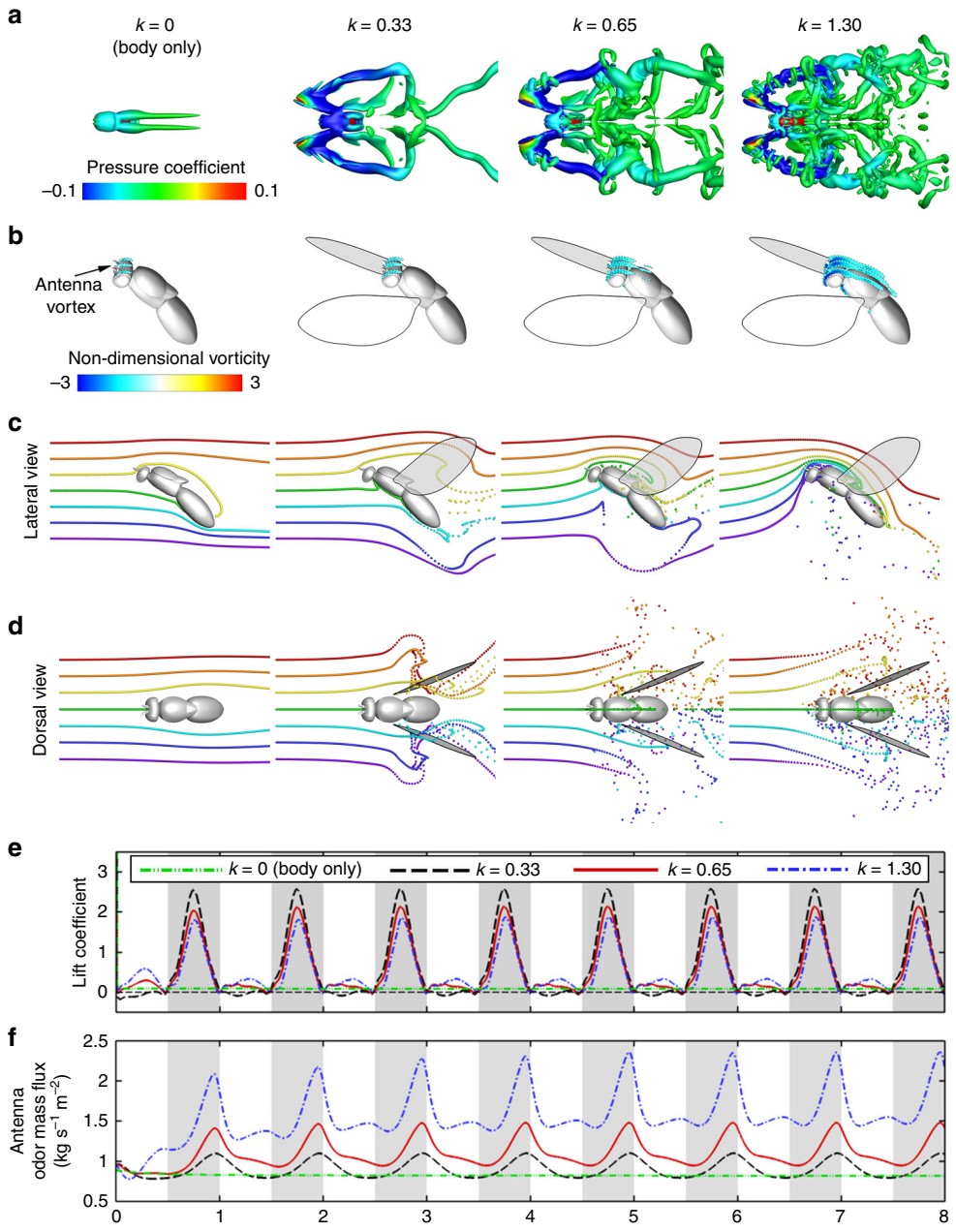

**Fig. 2** Fruit fly in forward flight at four different reduced frequencies. **a** Top view of the wake structures by Q-criterion and color coded with normalized pressure. **b** Vortex structures over antennae of the fruit fly and visualized using span-wise vorticity. Blue indicates clockwise vorticity. Both wake structure and antenna vortex intensify at higher reduced frequencies. **c**, **d** Odor plume structure visualized using neutral-buoyant particles from lateral (**c**) and dorsal (**d**) releases: snapshot at the time point of peak odor mass flux to antennae. The colors of the particle indicate different release locations. As flapping frequency increases, more odor particles vertically below the body are trapped and flicked up toward the antennae. However, horizontally, only a narrow stream of particles (green) in the direct path of antenna passes through the antenna region, regardless of flapping frequency (see also Supplementary Movie 1). **e**, **f** Time course of lift coefficient (**e**) and odor mass flux over the antennae (**f**) under different reduced frequency. The shaded areas represent downstrokes. Lift coefficients peak at mid-downstroke phase and decrease with higher reduced frequencies. However, antenna odor mass flux peaks during the downstroke to upstroke transition phase and increases with higher reduced frequencies

same trend. The lift coefficient ($C_L$) is the total lift force non-dimensionalized by wing-tip velocity squared and wing area, which generally reflects the efficiency of wing shape and design[27]. Higher lift coefficients can translate into carrying more payload per unit wing area[28]. The data show that increased reduced frequency (from $k = 0.65$ to $k = 1.30$) enhances peak odor mass flux over the antennae (59%) but slightly decreases the lift coefficient (−11%). More importantly, the cycle-average lift coefficient

distribution contour plot on the wing surface (Fig. 3a; Supplementary Fig. 2) shows that the trailing-edge portions that are important to odor transport as observed previously (upward flicking) contribute poorly to lift generation.

Intrigued by this observation, we virtually cut off the trailing-edge portion of the fruit fly wing (Fig. 3b) and reran the flight simulation while maintaining all other settings the same. Lift production (Fig. 3a–c), vortex formation (Fig. 3d–f), and odor

| Table 1 Aerodynamic performance and odor mass flux over the antennae at various reduced frequencies | | | | | | |
|---|---|---|---|---|---|---|
| Reduced frequency ($k=fR/U_\infty$) | Lift coefficient ($\overline{C}_L$) | Total force coefficient ($\overline{C}_F$) | Aerodynamic power coefficient ($\overline{C}_{PW}$) | Total force-to-power ratio ($\overline{C}_F/\overline{C}_{PW}$) | Peak odor mass flux (kg s$^{-1}$ m$^{-2}$) | Mean odor mass flux (kg s$^{-1}$ m$^{-2}$) |
| 0 (body only) | 0.02 | 0.05 | 0.09 | 0.56 | 0.82 | 0.82 |
| 0.33 | 0.62 | 0.81 | 0.51 | 1.59 | 1.10 | 0.91 |
| 0.43 | 0.60 | 0.76 | 0.42 | 1.82 | 1.22 | 0.97 |
| 0.65 | 0.57 | 0.72 | 0.36 | 2.01 | 1.48 | 1.12 |
| 0.97 | 0.53 | 0.68 | 0.34 | 2.00 | 1.92 | 1.40 |
| 1.30 | 0.51 | 0.66 | 0.33 | 1.99 | 2.35 | 1.67 |

The average bar ("−") indicates the value averaged over a flapping cycle

transport (Fig. 3g–i) were compared side by side over a wide range of reduced frequencies (Fig. 3j–l). The modified wing improves the average lift coefficient by 9.6% at $k = 0.65$ and by 18.0% at $k = 1.30$, as well as improving the overall aerodynamic efficiency, evaluated using the ratio of total force generated (combining both lift and forward thrust force) over total power consumed (Fig. 3k: by 4.3% at $k = 0.65$ and by 6.3% at $k = 1.30$). The reason: the trailing-edge portion of the wing accounts for ~20% of wing area yet accounts for only <5% of lift force generation (Supplementary Fig. 4); thus, removing it significantly reduces the power needed to flap the wing against air resistance while improving the lift coefficient as well as overall aerodynamic power economy. However, this wing modification results in a significant reduction of peak odor mass flux over the antennae, by −10.7% at $k = 0.65$ and by −17.9% at $k = 1.30$ (Fig. 3l). The modification does not significantly affect the leading-edge vortex formation and circulation of the wing but significantly reduces the strength of the vortex around the antenna (Fig. 3d–f), which explains the differential impact on aerodynamic performance versus antenna odor flux. In addition, the vertical spatial sampling range was also decreased compared to the original wing (Fig. 3g, h; Supplementary Movie 1, 02:51–04:07). For example, after the wing shape modification, the blue and purple particles cannot be pushed over the antenna region at $k = 1.30$ (Supplementary Movie 1, 03:29-04:07). This wing manipulation confirms that a wider trailing edge leads to a stronger odor-trapping and flicking effect during flight and suggests that the original wing shape may not be optimal for aerodynamic performance but may result in better olfactory performance.

## Discussion

Some 400 million years ago, insects evolved wings and the ability to fly[29]. Flight allows them to escape from ground predators, to explore farther for food sources and mates, and to fill new ecological niches that ground animals cannot reach[30]. With this critical advantage, insects quickly become the most diverse and abundant animal group[7]. Through natural selection, insects have developed very complicated wing designs and flying mechanisms to enhance aerodynamic performance that is still beyond our completely knowledge, including delayed stall[13], wake capture[13], clip and fling[17], trailing-edge vortices[14], wing–wing interactions[31], wing–body interactions[19], and etc. There is a common belief that insect wings have evolved to be highly aerodynamically efficient[32,33] and that even slight changes in wing geometry or flapping kinematics could lead to loss in aerodynamic performance[34–36]. Yet, a different challenge arises when insects take to the air: their wing flapping now inevitably perturbs incoming chemosensory cues. How do insects address the conflict between aerodynamic performance and olfactory function? Our study, through the use of computational fluid dynamics simulations, quantitatively confirms and clarifies that flapping wings may

enhance olfactory stimuli to the perfectly positioned primary olfactory organs (antennae) and offers new insights that (1) because both aerodynamic and olfactory functions are indispensable during odor-guided navigation, some aerodynamic performance may be sacrificed to improve olfactory performance, and (2) the shape and size of the wing may be a balance between the two functions. The wide trailing edge close to the wing root might not be optimal in terms of aerodynamics, but it can induce strong airflow over insect antennae. Furthermore, we found that higher flapping frequencies and strong wing transition phases induced higher odor mass flux, while lower flapping frequencies and downstroke phases produce better lift coefficients—again, a balance between the two functions.

The seemingly effortless flying and odor-tracking abilities of many insects remarkable for their tiny size have captivated the interest of naturalists and biologists for centuries. Insect wings are a remarkable evolutionary product that are known to serve diverse roles in addition to flying, including pheromone dispersal[37], sound production[38,39], and ventilation of hives[40,41]. *Drosophila* may use their wings in a courtship display for engaging potential mates[42]. Beetles have evolved a hardened forewing (or elytron) as armor protection[43], as well as for improving aerodynamic performance by interacting with the flapping hindwings during flight[44]. The results of our study critically expand the basic understanding of insect wing functions and insect biology by further revealing that optimal aerodynamics may be traded for more efficient olfactory performance during flight and may inspire future novel biological and neuroethology investigations. Directly assessing the impact of wing and antenna geometries, kinematics, and spatial orientations on olfactory sensitivities for different species of insects may help elucidate characteristics that improve odor-guided navigation. The interaction between wing flapping, the anisotropic spatial sampling ranges, and complicated incoming odor plume structure that insects might experience in the field may have further implications in understanding the behavior and algorithm of plume tracking of insects. The findings can also contribute to the design of future, more efficient micro-aerial vehicles with onboard chemical detectors.

## Methods

**Model fruit fly**. A morphological-accurate model of the fruit fly *D. melanogaster* was constructed (Fig. 1a; Supplementary Fig. 1a). The wing shape was digitized from a *D. melanogaster* wing[45] (Supplementary Fig. 1b) that has a wing area of 2.59 mm$^2$, a wingspan ($R$) of 2.87 mm, and an average chord ($\bar{c}$) of 0.89 mm. Left–right symmetry was assumed. The fruit fly wings, small and relatively stiff, are assumed to be rigid during flapping motion based on previous literature[46,47].

Based on the previous literature on forward-flying insects[22,23,48], wing kinematics were prescribed with sinusoidal function of wing position angle $\phi(t) = 0.5\Phi\cos(2\pi ft)$ with an amplitude of $\Phi = 140°$, setting the wing deviation angle $\theta = 0°$ with respect to the stroke plane, and assuming a constant wing feathering angle $\alpha$ of 60° during the upstroke and −30° during the downstroke. At the ventral and dorsal stroke reversal, $\alpha$ changed sinusoidally over the duration of the 0.22-cycle period. This wing motion presents as an idealized flapping motion used by insects during a forward flight motion. The wing Euler angle profiles are illustrated in Supplementary Fig. 1c, and the wing chord kinematics are visualized in

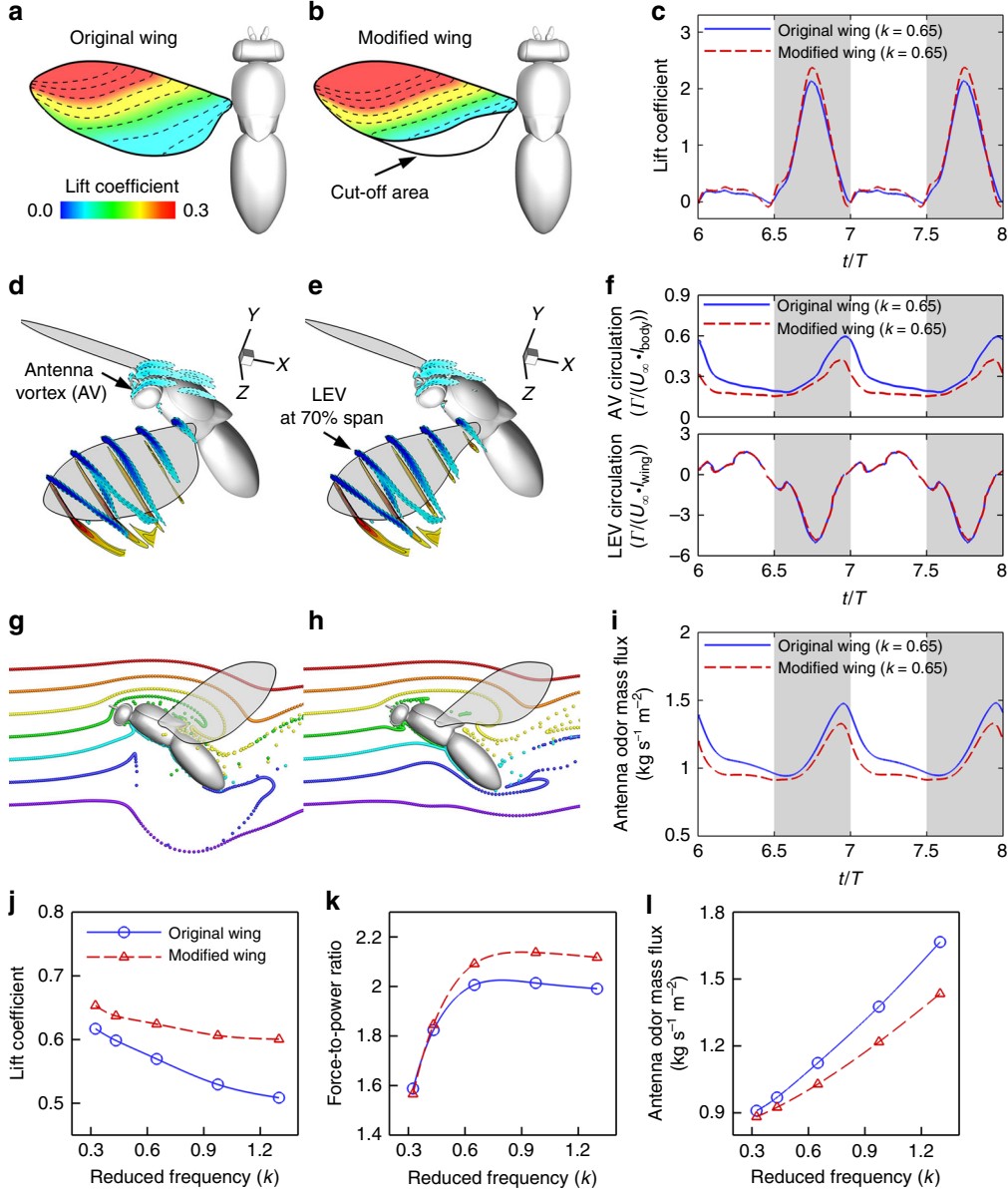

**Fig. 3** Side-by-side comparisons between the original and modified wings. **a**, **b** Comparison between original and modified wings that cut off part of the trailing edge at $k = 0.65$. Color contour indicates the cycle-averaged lift coefficient on the wing surface. **c** Time course of lift coefficient. **d**, **e** Comparison of antenna vortex (AV) and leading-edge vortex (LEV) formation at the mid-downstroke. **f** Time course of vortex circulation of AV at the body center and LEV at 70% wingspan. **g**, **h** Odor plume structures visualized by neutral-buoyant particles. (**i**) Time course of odor mass flux over antennae. **j–l** Cycle-averaged lift coefficient (**j**), total force-to-power ratio (**k**), and odor mass flux at antenna (**l**) as function of the reduced frequency (**k**). The modified wings produced similar LEV (**f**, bottom plot), better lift coefficient (**c**, **j**), better force-to-power ratio (**k**), but significantly worse antenna vortex (**f**, top plot) and odor mass flux (**i**, **l**). **a**, **b** Comparison between original and modified wings that cut off part of the trailing edge at $k = 0.65$. Color contour indicates the cycle-averaged lift coefficient on the wing surface. **c** Time course of lift coefficient. **d**, **e** Comparison of antenna vortex (AV) and leading-edge vortex (LEV) formation at the mid-downstroke. **f** Time course of vortex circulation of AV at the body center and LEV at 70% wingspan. **g**, **h** Odor plume structures visualized by neutral-buoyant particles. **i** Time course of odor mass flux over antennae. **j–l** Cycle-averaged lift coefficient (**j**), total force-to-power ratio (**k**), and odor mass flux at antenna (**l**) as function of the reduced frequency (**k**). The modified wings produced similar LEV (**f**, bottom plot), better lift coefficient (**c**, **j**), better force-to-power ratio (**k**), but significantly worse AV(**f**, top plot) and odor mass flux (**i**, **l**)

Supplementary Fig. 1d. The inclination angle of the stroke plane ($\beta$) against the horizontal body axis is 20°. The entire body is inclined by $\chi = 45°$ with respect to the horizontal plane.

The clipping of wing trailing edge discussed in the main text is based on the surface contour of the cycle-averaged lift coefficient of the original wing at $k = 0.65$ (Supplementary Fig. 5). The modified wing area and mean chord length are 2.09 mm$^2$ and 0.73 mm, respectively.

**Numerical method**. The numerical simulations were performed using a second-order Cartesian grid-based immersed boundary method. The details of this solver have been previously described;[24] brief descriptions are provided here.

The non-dimensional equations governing the flow in the numerical solver were the time-dependent viscous incompressible Navier–Stokes equations, written in indicial form, as follows:

$$\frac{\partial u_i}{\partial t} + \frac{\partial (u_i u_j)}{\partial x_j} = -\frac{\partial p}{\partial x_i} + \frac{1}{Re}\frac{\partial}{\partial x_j}\left(\frac{\partial u_i}{\partial x_j}\right) \tag{1}$$

$$\frac{\partial u_i}{\partial x_i} = 0 \tag{2}$$

where $u_i$ ($i = 1,2,3$) are the velocity components in the $x$-, $y$-, and $z$-directions, respectively; $p$ is the pressure; and Re is the Reynolds number.

Equations 1 and 2 were discretized using a second-order central difference scheme on a nonuniform Cartesian mesh, where the velocity and pressure are collocated at the cell centers. The unsteady equations were solved using a fractional step method, which provides second-order accuracy in time. An Adams–Bashforth scheme and an implicit Crank–Nicolson scheme were used to discretize the convective terms and diffusion terms, respectively. Boundary conditions on immersed bodies were imposed through a "ghost-cell" procedure, and the flow simulations were conducted on stationary non-body-conformal Cartesian grids. This arrangement eliminates the complicated remeshing algorithms usually needed for conventional Lagrangian body-conformal methods.

**Simulation setup.** Simulations were conducted on a nonuniform 289×137×249 (about 10 million)-point Cartesian grid. The overall computational domain had dimensions of a 15R×15R×15R cubic box, where $R = 2.87$ mm is the wingspan length. To resolve the near-wake vortex structures, a cuboidal area around the fruit fly with dimensions of $2R×1R×2.5R$ had a high-resolution uniform grid ($\Delta \cong 0.0125R$), as shown in Supplementary Fig. 6a. Stretching grids were applied in all three directions from the fine region to the outer boundaries. At the left-hand boundary, a constant inflow velocity boundary condition is applied. The right-hand boundary is the outflow boundary, where a zero stream-wise gradient boundary condition was applied for the velocity, allowing the vortices to convect out of this boundary without significant reflections. The zero-stress boundary condition was applied at all lateral boundaries. A homogeneous Neumann boundary condition was provided for the pressure at all boundaries. High-density triangular surface mesh specify the surface of the fruit fly's body and wings (see Supplementary Fig. 6b). Nonslip boundary conditions were applied on both body and wing surfaces. To guarantee that the entire flow field reached a periodic state[49,50], all simulations were run for eight flapping cycles. This running period also ensured that the wake structures generated by flapping wings were fully affected by the outflow boundary condition.

Grid refinement was performed to ensure that the simulation results were grid independent. Supplementary Fig. 7 presents the comparison of lift and forward thrust coefficients in three different girds densities. The plots show that the differences between the medium grid (presented in this article) and fine grid are <2.1% for lift coefficient and 0.9% for thrust coefficient, at their peaks. This demonstrates that the results of the current study are grid independent. In addition, the thrust coefficient in Supplementary Fig. 7b has positive and negative values, indicating that the fruit fly produced thrust in the upstroke and drag in the downstroke. The cycle-averaged thrust coefficient is close to zero (~0.018). Thus, the force balance is approximately achieved in the horizontal direction at $k = 0.65$, which is close to a self-propelled forward flight. At other flapping frequencies, the fruit fly should be considered as tethered.

The Reynolds number in forward flight is defined as $Re = U_\infty R/\nu$, where $U_\infty$ represents the forward flight speed (0.94 m/s) and $\nu$ is the kinematic viscosity ($1.56 × 10^{-5}$ m$^2$ s$^{-1}$) for air at room temperature (27 °C). Based on the definition, the Reynolds number in this study is 173. The reduced frequency is defined as $k = fR/U_\infty$, where $f$ is the flapping frequency. In the unsteady aerodynamics, the reduced frequency is a dimensionless number that used to define the degree of unsteadiness of the flow filed. To change the reduced frequency, we can either adjust the forward flight speed ($U_\infty$) or adjust the wing flapping frequency ($f$), with the similar effect on aerodynamics. (see Supplementary Fig. 2). Thus, in the current study, we used a single incoming flow velocity and varied the wingbeat frequencies (summarizes Supplementary Table 1). The results may be extrapolated to other forward speeds based on matching Re and reduced frequencies.

**Evaluation of the aerodynamic force and power.** The instantaneous aerodynamic forces acting on the wing surface can be calculated from the pressure and stresses along its surface based on the solutions to the Navier–Stokes equations. The lift and thrust force ($F_L$, along the vertical direction; $F_T$ along the horizontal direction) are presented as non-dimensional lift and thrust coefficients, which are computed by $C_{L,T} = (F_L, F_T)/0.5\rho \bar{U}_{tip}^2 S$, where $C_L$ and $C_T$ are the lift and thrust coefficients and $S$ is the area of the wing surface. $\bar{U}_{tip}$ is the mean wing-tip velocity, defined as $\bar{U}_{tip} = (1/T)\int_0^T \sqrt{u_{tip}^2 + v_{tip}^2 + w_{tip}^2}\,dt$, where $u_{tip}, v_{tip}$, and $w_{tip}$ are wing-tip velocity components in $x$-, $y$-, and $z$-directions, respectively. Similarly, the non-dimensional total force coefficient is given by $C_F = F_{total}/0.5\rho \bar{U}_{total}^2 S$, where the $F_{total}$ represents the total aerodynamic force generated by the wing, a combination of both lift and thrust forces. The aerodynamic power consumption ($P_{aero}$) is the power needed to flap the wing against air resistance. The non-dimensional aerodynamic power coefficient is defined as $C_{PW} = P_{aero}/0.5\rho \bar{U}_{tip}^3 S$, same as previous studies on fruit fly[51], cicada[19], and dragonfly[52] flight. The overall aerodynamic efficiency is evaluated using the ratio of total force generated over total power consumed, which is defined as $C_F/C_{PW}$.

Supplementary Fig. 4 compares the lift coefficient and lift force generated by the original wing and the modified wing. The trailing-edge region accounts for ~20% of the total wing area yet only contributes <5% of lift generation over all frequencies. Thus, its removal improves lift coefficient as well as the overall aerodynamic efficiency (force-to-power ratio) (Fig. 3k), due to less power consumed to flap the wing against air resistance.

**Validation of numerical method.** To validate the numerical method used in the present study, a separate numerical simulation of the fruit fly was conducted to replicate experiments of Sane and Dickinson[53]. The wing sweeps in the horizontal plane and rotates at the end of each stroke. The stroke amplitude was 180°, and the angle of attack at the midstroke was 50°. The Reynolds number was 136. A non-uniform Cartesian grid of size 256×144×192 was used in a computational domain of $30\bar{c}× 30\bar{c}× 30\bar{c}$ to obtain domain-independent results. The comparisons to the experimental measurements[53] and previous numerical simulations[22,54] are shown in Supplementary Fig. 8. The magnitude and variation of the computed lift and drag forces agree reasonably well with the previous results.

**Quantification of odor mass flux around antennae.** The governing equation of odorant convection and diffusion in the air phase is

$$\frac{\partial C'}{\partial t} = D\frac{\partial^2 C'}{\partial x_i \partial x_i} - u_i\frac{\partial C'}{\partial x_i} \qquad (3)$$

where $i = 1,2,3$ indicate the components in the $x$-, $y$-, and $z$-directions; $C'$ is the normalized odorant concentration defined by $C' = C/C_{in}$, in which $C_{in}$ is the inlet or ambient air odorant concentration ($C'$ at the inlet boundary equals 1). The normalized uniform inlet concentration allows us to focus on the effect of wing flapping. In the future, the more complicated odor plume structure that the insect might experience in the field can be introduced.

The Peclet number for the mass transfer is defined by $Pe = Re\ Sc$, where Sc represents the Schmidt number, which is the ratio between kinematic viscosity and mass diffusivity ($Sc = \nu/D$). Typical natural odor in the environment has quite low diffusivity ($D$) in air at normal temperature and pressure, ranging from $10^{-1}$ to $10^{-2}$ cm$^2$/s. Thus, the Sc has a range of $10^0$ to $10^1$. Based on the definition, the Peclet number in the current study is $10^2$ to $10^3$; thus, convective transport due to air movement dominates the system for most natural odors, and odor diffusion may be ignored. By ignoring odorant diffusion, the first term in the right-hand side of Eq. 3 is treated as zero. The odor mass flux over antennae is then calculated as $C'\ \rho_{odor}\ U^*$, where $\rho_{odor}$ is the density of odor and $U^*$ represents the air velocity at 0.03R above the antenna surface (since air velocity is always zero at the surface). So this equation assumes 100% odor absorption when the odorant-laden air passes through the olfactory organ, which is a reasonable simplification for an initial study based on Sc number. Sc number is the ratio of momentum diffusivity and mass diffusivity, and is higher than 1 for most common odors. The diffusion and binding process of specific odor through the boundary layer to the olfactory structure and to olfactory receptor will certainly be worthy of further investigation in the future. The instantaneous profiles of $U^*$ are obtained by averaging three virtual probes around the antenna. In addition, the density ratio between odor particles and air is assumed to be 1 ($\rho_{odor} = \rho_{air} = 1.225$ kg m$^{-3}$). Similar probes are placed at 10 different locations 0.03R above the body surface around fruit fly body (see Supplementary Fig. 3).

**Lagrangian tracking of odor structures.** To visualize the odor plume structures, the Lagrangian tracking approach is applied by assuming that odor transport is dominated by the convective flow field, as described above. Computational neutral-buoyant particle tracers have been widely used to mimic smoke[55] and bubbles[56] when diffusion is low, with good experimental agreement. The time step was set as 0.001 s.

**Code availability.** The in-house CFD solver algorithm[24] has been published elsewhere. The executable file of the code is available from the authors upon reasonable request for non-commercial purposes only.

**Data availability.** Data that support the findings of this study are available from the authors upon reasonable request.

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

## Acknowledgements

This research is supported by NIH NIDCD R01 DC013626 to K.Z. and NSF CBET-1313217 to H.D.

## Author contributions

K.Z. conceived the idea; C.L. and K.Z. designed the study; C.L. carried out the computational simulations; C.L., K.Z., and H.D. analyzed the data; K.Z., C.L., and H.D. wrote the manuscript.

## Additional information

**Competing interests:** The authors declare no competing interests.

