## [Peer Review File · Nature Communications]

REVIEWERS' COMMENTS:

Reviewer #1 (Remarks to the Author):

The authors have done a great job addressing most of my concerns from the previous review, and the paper is much improved. I have some additional minor comments below.

Introduction

The revised introduction does a much better job motivating the study. I still think that the introduction is a bit too short given the breadth of the topic (aerodynamics, flight kinematics, and olfaction). I recommend dividing the introduction up into more than one paragraph and providing some additional background on flight aerodynamics so that the general reader can better understand the trade-offs. Also the reader should be orientated with respect to the scale of the problem. The fruit fly flies at Reynolds numbers on the order of 100 where turbulence at the scale of the antennae / body / wings is likely not relevant. Boundary layers can be relatively large and important for olfaction, etc.

Results

Given the broad readership of nature, when the Reynolds number is introduced the reader should be pointed towards its definition. Its significance should be explained in terms of both aerodynamics and olfaction. Consider simply stating in the main results that the Reynolds number describes the ratio of inertial to viscous forces in a fluid.

"This finding confirmed that the conventional wisdom is correct: during forward flight, the antenna is well positioned to receive significant increase of odor mass flux while avoiding significant turbulence compared to other locations along the body." Can you provide a reference for the conventional wisdom? Also, is the flow near the body really turbulent at $Re \sim 200$?

Discussion

Given the large body of work on the competing functions of wings in general (ventilation of hives in the case of bees, protection, courtship displays, etc), I would put this study into the larger context of this work.

Dual functions of insects wings: balancing aerodynamics and olfaction

We wish to express our sincere appreciation to the editor and the reviewer for their insightful comments, which assisted us in substantially improving our paper. The comments raised by the reviewer have been incorporated into the revised paper. Detailed responses are in the following.

Reply to Reviewer # 1:

The authors have done a great job addressing most of my concerns from the previous review, and the paper is much improved. I have some additional minor comments below.

Introduction

The revised introduction does a much better job motivating the study. I still think that the introduction is a bit too short given the breadth of the topic (aerodynamics, flight kinematics, and olfaction). I recommend dividing the introduction up into more than one paragraph and providing some additional background on flight aerodynamics so that the general reader can better understand the trade-offs. Also the reader should be orientated with respect to the scale of the problem. The fruit fly flies at Reynolds numbers on the order of 100 where turbulence at the scale of the antennae / body / wings is likely not relevant. Boundary layers can be relatively large and important for olfaction, etc.

Reply:

We thank the reviewer for his/her good suggestion. We have expanded our introduction section to provide a more comprehensive review regarding the aerodynamic and olfactory functions of the insect wings.

Also, we replaced the term “turbulence” with “vortex”, “air disturbance”, and “perturbation”, which are better descriptions of the situation.

Revised submission, page 1-2:

“

Insects have remarkable flying and odor-tracking capabilities¹⁻³ that have captivated the interest of naturalists and biologists for centuries. For example, a male moth can track its female counterpart from miles away via pheromone detection^{4, 5}. Insects possess a sophisticated olfactory system that is extremely sensitive to a great number of volatile chemicals, with its total detection capacity that has never been fully catalogued^{6, 7}. But then, how are olfactory detection and tracking affected by wing flapping during the flight, which inevitably perturbs the incoming odor plume? One widely held hypothesis is that flapping wings may actively draw odor plumes to the primary olfactory sensory organ, the antennae—an action analogous to “sniffing” in mammals, and that wing beating may be a critical part of active olfactory sampling for insects^{8, 9}. For example, a silkworm with its wings removed is unable to track odor plumes, even though it tracks plumes while walking¹⁰. Experimental measurements using hot-wire anemometers showed that the induced airflow generated by the wings varied with wingbeat frequency, which may alter olfactory stimuli to the sensory organs¹¹. Yet, there is a lack of quantitative details and confirmation on how flapping wings actively draw odor plumes to the antennae.

On the other hand, through million years of evolution, insects have developed superior wing designs and complex flying mechanisms to enhance their aerodynamic performance¹²⁻¹⁴. Wing flapping back and forth generates a tornado-shaped leading-edge vortex on its top

surface that can provide almost twofold lift compared with static wings^{15,16}. Unsteady wing flapping motion can augment lift through delayed stall when the wing sweeps through the air with high angle of attack during the translational phase. Rapid wing reversal can capture its own wake and thus further enhance lift. Besides these three most common unsteady aerodynamic mechanisms¹³ (leading edge vortex, delayed stall, and wake capture), many insects also apply other unique aerodynamic mechanisms to further increase lift. For example, the slender wings of mosquitoes can take advantage of the trailing-edge vortex¹⁴. For insects like butterflies, their wings may interact with each other to enhance force generation through clap-and-fling and clap-and-peel aerodynamics^{17, 18}. For insects with wider body shape, such as cicadas, wing-body interaction mechanisms may play a critical role to enhance lift generation¹⁹.
”

Results

Given the broad readership of nature, when the Reynolds number is introduced the reader should be pointed towards its definition. Its significance should be explained in terms of both aerodynamics and olfaction. Consider simply stating in the main results that the Reynolds number describes the ratio of inertial to viscous forces in a fluid.

“This finding confirmed that the conventional wisdom is correct: during forward flight, the antenna is well positioned to receive significant increase of odor mass flux while avoiding significant turbulence compared to other locations along the body.” Can you provide a reference for the conventional wisdom? Also, is the flow near the body really turbulent at $Re \sim 200$?

Reply:

We have added the physical meaning of the Reynolds number where the term is first introduced to the readers. We also added a reference for supporting the conventional wisdom.

Thank you for pointing out the misuse of the term “turbulence” in our manuscript. The air vortex generated by wing flapping at Reynolds number around 200 is still laminar flow. We have replaced the term by disturbance air disturbance or perturbation.

Discussion

Given the large body of work on the competing functions of wings in general (ventilation of hives in the case of bees, protection, courtship displays, etc), I would put this study into the larger context of this work..

Reply:

We thank the good comments from the reviewer. We have added a more comprehensive discussion regarding the diverse-functions of the flapping wings in our discussion section.

Revised submission, page 4:

“

Some 400 million years ago, insects evolved wings and the ability to fly²⁰. Flight allows them to escape from ground predators, to explore farther for food sources and mates, and to fill new ecological niches that ground animals cannot reach²¹. With this critical advantage, insects quickly become the most diverse and abundant animal group⁷.

”

Revised submission, page 5:

“

The seemingly effortless flying and odor-tracking abilities of many insects remarkable for their tiny size have captivated the interest of naturalists and biologists for centuries. Insect wings are a remarkable evolutionary product that are known to serve diverse roles in addition to flying, including pheromone dispersal²², sound production^{23,24}, and ventilation of hives^{25,26}. *Drosophila* may use their wings in a courtship display for engaging potential mates²⁷. Beetles have evolved a hardened forewing (or elytron) as armor protection²⁸, as well as for improving aerodynamic performance by interacting with the flapping hindwings during flight²⁹. The results of our study critically expand the basic understanding of insect wing functions and insect biology by further revealing that optimal aerodynamics may be traded for more efficient olfactory performance during flight and may inspire future novel biological and neuroethology investigations.

”

References

1. van Breugel F, Dickinson MH. Plume-tracking behavior of flying *Drosophila* emerges from a set of distinct sensory-motor reflexes. *Current Biology* **24**, 274-286 (2014).
2. Daly KC, Kalwar F, Hatfield M, Staudacher E, Bradley SP. Odor detection in *Manduca sexta* is optimized when odor stimuli are pulsed at a frequency matching the wing beat during flight. *PLoS One* **8**, e81863 (2013).
3. Manar F, Medina A, Jones AR. Tip vortex structure and aerodynamic loading on rotating wings in confined spaces. *Experiments in fluids* **55**, 1815 (2014).
4. Wall C, Perry J. Range of action of moth sex - attractant sources. *Entomologia experimentalis et applicata* **44**, 5-14 (1987).
5. Collins CW, Potts SF. *Attractants for the flying gipsy moths as an aid in locating new infestations*. US Department of Agriculture (1932).
6. Laissue PP, Vosshall LB. The olfactory sensory map in *Drosophila*. In: *Brain development in Drosophila melanogaster*. Springer (2008).
7. Szyszka P, Galizia CG. Olfaction in insects. *Handbook of Olfaction and Gustation*, 531-546 (2015).
8. Tripathy SJ, Peters OJ, Staudacher EM, Kalwar FR, Hatfield MN, Daly KC. Odors pulsed at wing beat frequencies are tracked by primary olfactory networks and enhance odor detection. *Frontiers in cellular neuroscience* **4**, (2010).

9. Loudon C, Koehl M. Sniffing by a silkworm moth: wing fanning enhances air penetration through and pheromone interception by antennae. *Journal of experimental Biology* **203**, 2977-2990 (2000).
10. OBARA Y. Bombyx mori Mating Dance: an Essential in Locationg the Female. *Applied Entomology and Zoology* **14**, 130-132 (1979).
11. Sane SP, Jacobson NP. Induced airflow in flying insects II. Measurement of induced flow. *J Exp Biol* **209**, 43-56 (2006).
12. Ellington CP, vandenBerg C, Willmott AP, Thomas ALR. Leading-edge vortices in insect flight. *Nature* **384**, 626-630 (1996).
13. Dickinson MH, Lehmann FO, Sane SP. Wing rotation and the aerodynamic basis of insect flight. *Science* **284**, 1954-1960 (1999).
14. Bomphrey RJ, Nakata T, Phillips N, Walker SM. Smart wing rotation and trailing-edge vortices enable high frequency mosquito flight. *Nature* **544**, 92 (2017).
15. van den Berg C, Ellington CP. The vortex wake of a 'hovering' model hawkmoth. *Philosophical Transactions of the Royal Society of London Series B: Biological Sciences* **352**, 317-328 (1997).
16. Birch JM, Dickinson MH. Spanwise flow and the attachment of the leading-edge vortex on insect wings. *Nature* **412**, 729-733 (2001).
17. Weis-Fogh T. Quick Estimates of Flight Fitness in Hovering Animals, Including Novel Mechanisms for Lift Production. *Journal of Experimental Biology* **59**, 169-230 (1973).
18. Lehmann FO. When wings touch wakes: understanding locomotor force control by wake-wing interference in insect wings. *Journal of Experimental Biology* **211**, 224-233 (2008).
19. Liu G, Dong H, Li C. Vortex dynamics and new lift enhancement mechanism of wing-body interaction in insect forward flight. *Journal of Fluid Mechanics* **795**, 634-651 (2016).
20. Whiting MF, Bradler S, Maxwell T. Loss and recovery of wings in stick insects. *Nature* **421**, 264 (2003).
21. Misof B, *et al.* Phylogenomics resolves the timing and pattern of insect evolution. *Science* **346**, 763-767 (2014).

22. Peters JM, Gravish N, Combes SA. Wings as impellers: honey bees co-opt flight system to induce nest ventilation and disperse pheromones. *Journal of Experimental Biology* **220**, 2203-2209 (2017).
23. SPANGLER HG, GREENFIELD MD, TAKESSIAN A. Ultrasonic mate calling in the lesser wax moth. *Physiological Entomology* **9**, 87-95 (1984).
24. Alexander RD. Sound production and associated behavior in insects. (1957).
25. Southwick EE, Moritz RF. Social control of air ventilation in colonies of honey bees, *Apis mellifera*. *Journal of insect physiology* **33**, 623-626 (1987).
26. Seeley TD. Atmospheric carbon dioxide regulation in honey-bee (*Apis mellifera*) colonies. *Journal of Insect Physiology* **20**, 2301-2305 (1974).
27. Spieth HT. Courtship behavior in *Drosophila*. *Annual review of entomology* **19**, 385-405 (1974).
28. Pearson DL. Biology of tiger beetles. *Annual Review of Entomology* **33**, 123-147 (1988).
29. Le TQ, *et al.* Improvement of the aerodynamic performance by wing flexibility and elytra–hind wing interaction of a beetle during forward flight. *Journal of The Royal Society Interface* **10**, 20130312 (2013).